# Research on China’s Risk of Housing Price Contagion Based on Multilayer Networks

**DOI:** 10.3390/e24091305

**Published:** 2022-09-15

**Authors:** Lu Qiu, Rongpei Su, Zhouwei Wang

**Affiliations:** School of Finance and Business, Shanghai Normal University, Shanghai 200234, China

**Keywords:** housing price contagion risk, multilayer networks, transfer entropy, generalized variance decomposition

## Abstract

The major issue in the evolution of housing prices is risk of housing price contagion. To model this issue, we constructed housing multilayer networks using transfer entropy, generalized variance decomposition, directed minimum spanning trees, and directed planar maximally filtered graph methods, as well as China’s comprehensive indices of housing price and urban real housing prices from 2012 to 2021. The results of our housing multilayer networks show that the topological indices (degree, PageRank, eigenvector, etc.) of new first-tier cities (Tianjin, Qingdao, and Shenyang) rank higher than those of conventional first-tier cities (Beijing, Shanghai, Guangzhou, and Shenzheng).

## 1. Introduction

Owing to the rapid development of China’s real estate market, housing is a source of concern for ordinary residents and investors, whether for living or investment purposes. Moreover, the factors that influence and cause changes in housing prices are a hot topic. As information exchange becomes increasingly common, obtaining housing price information from various regions has become easier, potentially leading to a correlation effect between housing prices in different regions. Changes in housing prices in one region may cause changes in housing prices in other regions. Therefore, it is necessary to study the relationship between housing prices in different regions to avoid the risk of contagion between regions when housing prices in one place collapse and cause real estate in most regions to experience a downturn through the aforementioned linkage. Moreover, such investigations can improve the effectiveness of housing price control measures and provide useful advice with respect to reducing the systemic risk of real estate.

Research on the correlation of housing prices between cities can be divided into two categories: studies related to the spillover correlation of housing prices using spatial measurement methods and those that investigate the correlation between sequences using time series (Li Zheng et al., 2021) [1]. In general, for spatial econometric models, researchers use Moran’s I index to test the spatial correlation between housing prices before using spatial panel models to measure the spatial spillover effect of housing prices between cities. Ding et al. (2015) [2] used spatial econometrics to compute spatial correlations between 288 cities in China at the prefecture level and above. Based on spatial adjacency, spatial distance, economic adjacency, and economic distance matrices, Chen et al. (2012) [3] investigated the regional interaction of housing prices in China. Gong et al. (2020) [4] constructed an adjacency matrix based on distances and used the spatial lag of an X model (SLX), a spatial Durbin model, and a spatial Durbin error model to assess the spillover effect of housing prices in the Chinese regions of Jiangsu, Zhejiang, Shanghai, and Anhui.

The aforementioned studies revealed a positive spatial network spillover effect related to housing prices. Time-series analysis methods are more prevalent in housing price linkage research than spatial econometric models. Early research focused primarily on the “ripple effect”, which posits that housing prices have the characteristics of continuous transmission in space. Using data on British urban housing prices, MacDonald et al. (1993) [5] and Alexander (1994) [6] confirmed the existence of a ripple effect. As time-series analysis has become increasingly sophisticated, various methods for analyzing the correlation between series have enriched research on housing price correlations. To test the spatial relationship of urban housing prices, researchers have used the gravity model, the Granger causality test, the cointegration test, impulse response analysis, the generalized autoregressive conditional heteroskedasticity (GARCH) model, and other methods. Wang et al. (2015) [7] used the generalized impulse response of the vector error correction model to investigate the spillover effects of housing price bubbles in Beijing, Tianjin, Shanghai, and Chongqing and discovered that the degree of spillover effects and the direction of overflow differs among the four cities. Some researchers have combined the vector autoregression (VAR) models with the generalized variance decomposition (GVD) technique to study the spillover relationship of housing prices. For example, Yang et al. (2018) [8] combined principal component analysis with the GVD of the VAR models to study the effects of housing price spillovers in 69 large- and medium-sized Chinese cities. Moreover, Lv et al. (2019) [9] investigated the spillover effect of housing prices in 35 large- and medium-sized Chinese cities using the GVD of the thick-tailed VAR models.

In addition to using the VAR model, additional research has been conducted on the level of systemic risk in the real estate industry and the associated risk contagion following the subprime mortgage crisis. Li et al. (2019) [10] used the Tail Event driven NETwork (TENET) method to construct a systemic risk spillover network across industries. Liu et al. (2014) [11] used the AR-GARCH-CoVaR method to assess systemic risk spillovers in the real estate industry. The GARCH model has also been effectively used to evaluate the spillover contagion and correlation between housing prices and interest rates, exchange rates, and the stock market. Liu et al. (2016) [12] investigated the spillover and nonlinear correlation between housing prices, exchange rates, and stock prices using a smooth transition vector error correction GARCH (STVEC-GARCH) model. Yamaka et al. (2022) [13] investigated the nonlinear causality and dynamic correlation between exchange rates and housing prices in the boom-and-business market using the panel quantile Granger causality and dynamic conditional correlation (DCC) copula GARCH method. Some scholars have also applied the GARCH model to study the linkage effect of housing prices between regions. Zeng et al. (2015) [14] used the DCC-GARCH model to examine the relationship between housing price fluctuations in the three urban agglomerations of Beijing–Tianjin–Hebei, the Yangtze River Delta, and the Pearl River Delta. In research on the Granger causality test in housing price correlation networks, Chen et al. (2016) [15] evaluated the network structure characteristics of housing price linkage in 69 large- and medium-sized cities in China. They also extended the Granger causality test by including a nonlinear relationship to evaluate the correlation effect of housing prices in 70 large- and medium-sized Chinese cities [16].

The multilayer network method is rarely used to study the relationship between housing prices in the literature reviewed above. However, owing to the shortcomings caused by incomplete information from a single-layer network, we first constructed a multilayer network based on the comprehensive indices of housing prices (CIHP) and real housing prices of 31 major Chinese cities using the GVD and transfer entropy (TE) methods. Subsequently, we use the directed minimum spanning tree (DMST) and directed planar maximally filtered graph (DPMFG) methods to simplify each layer of the network and analyze the source and central cities in the housing price correlation network. The Multiplex Infomap method was used to conduct a community analysis of the multilayer network and study the geographical location and economic characteristics of cities with similar housing price fluctuation characteristics.

The remainder of this paper Is structured as follows. In Section 2, we introduce the theory of the TE and GVD networks, the calculation of multilayer network centralities, and Multiplex Infomap. In Section 3, we introduce the datasets and the CIHP construction, followed by presentation of the empirical results in Section 4. Finally, Section 5 comprises the conclusion and discussion.

## 2. Methodologies

### 2.1. Construction of Multilayer Networks

The TE method measures the transfer of housing price information between two cities, whereas the GVD method measures the impact of housing prices in other cities on housing prices in a given city. The correlation network created by these two methods is asymmetric, which corresponds to the asymmetric phenomenon of housing price fluctuation contagion in reality. Therefore, these two methods can describe the relationship between housing prices while also considerably reducing the bias that occurs in single-layer networks owing to the use of a multilayer network. Because the housing price correlation networks constructed by these two methods are all fully connected networks (except for the self-loop), it is difficult to grasp the key information. Consequently, we used the DMST and DPMFG methods to simplify the networks and build simplified multilayer housing price networks.

#### 2.1.1. Calculation of Transfer Entropy

According to Schreiber (2000) [17] and Chen et al. (2014) [18], we can calculate the transfer entropy from one city’s housing prices to those of another. A detailed introduction of the TE method can be found in Appendix A. Based on the TE method, the incidence matrix of housing prices can be calculated as follows:(1)$$TE\_W=\left[\begin{array}{ccccc} 0 & TE(1,2) & \cdots & TE(1,n-1) & TE(1,n)\\ TE(2,1) & 0 & \cdots & TE(2,n-1) & TE(2,n)\\ \vdots & \vdots & \ddots & \vdots & \vdots \\ TE(n-1,1) & TE(n-1,2) & \cdots & 0 & TE(n-1,n-1)\\ TE(n,1) & TE(n,2) & \cdots & TE(n,n-1) & 0 \end{array}\right]$$
where the TE TE(i,j) represents the information transfer from region I to region J. The diagonal element is set to 0 because there is no information transmission between housing prices in the same city.

#### 2.1.2. Calculation of Generalized Variance Decomposition

Based on Diebold and Yilmaz (2014) [19], we can construct the contagion matrix by the method of generalized variance decomposition; further details can be found in Appendix A. The elements of the GVD matrix for the H-step forecast are calculated as follows:(2)$$d_{ij}^{H}=\frac{\sigma_{ii}^{-1}\sum_{h=0}^{H-1}{\left(e_{i}^{\prime }A_{h}\Sigma e_{j}\right)}^{2}}{\sum_{h=0}^{H-1}\left(e_{i}^{\prime }A_{h}\Sigma A^{\prime }e_{i}\right)}$$
where $e_{j}$ is the unit vector, the element of which at j is 1 and the rest are 0; Σ denotes the covariance matrix of the random disturbance vector ($\varepsilon_{t}$); $\sigma_{ii}$ is the $\varepsilon_{t}$ standard deviation; H denotes the forecast period; and h is the disturbance in the moving average formula term lag order.

Because the row sum of the GVD is not always 1, the elements in the generalized variance matrix are standardized to better analyze the spillover relationship between housing prices and can be calculated as follows:(3)$${\tilde{d}}_{ij}=\frac{d_{ij}}{\sum_{j=1}^{N}d_{ij}}$$

Subsequently, the variance decomposition matrix can be obtained as:(4)$$D_{ij}(h)={\left(\begin{array}{llll} {\tilde{d}}_{11} & {\tilde{d}}_{12} & \cdots & {\tilde{d}}_{1N}\\ {\tilde{d}}_{21} & {\tilde{d}}_{22} & \cdots & {\tilde{d}}_{2N}\\ \vdots & \vdots & \ddots & \vdots \\ {\tilde{d}}_{N1} & {\tilde{d}}_{N2} & \cdots & {\tilde{d}}_{NN} \end{array}\right)}_{N\times N}$$

The diagonal of the obtained GVD matrix is set to 0; only the elements related to housing price overflow are retained. The processed variance decomposition matrix is:(5)$$VD\_W=\left[\begin{array}{ccccc} 0 & {\tilde{d}}_{1,2} & \cdots & {\tilde{d}}_{1,N-1} & {\tilde{d}}_{1,N}\\ {\tilde{d}}_{2,1} & 0 & \cdots & {\tilde{d}}_{2,N-1} & {\tilde{d}}_{2,N}\\ \vdots & \vdots & \ddots & \vdots & \vdots \\ {\tilde{d}}_{N-1,1} & {\tilde{d}}_{N-1,2} & \cdots & 0 & {\tilde{d}}_{N-1,N}\\ {\tilde{d}}_{N,1} & {\tilde{d}}_{N,2} & \cdots & {\tilde{d}}_{N,N-1} & 0 \end{array}\right]$$

#### 2.1.3. Steps of the DMST Method

The housing price correlation network built using the two aforementioned methods is fully connected and has a total of N(N-1) edges. However, the number of samples is too large to use a fully connected network to identify and analyze important nodes. Therefore, an appropriate method for simplifying the graph is required. Because the two correlation matrices constructed herein are both directed matrices, we simplify network graphs using a DMST (Kwon et al., 2008; Qiu et al., 2020) [20,21] and a DPMFG (Ye et al., 2019) [22]. The DMST procedure is as follows.
Randomly select a node as the root node;Travel all edges and find the smallest entry edges of all points except for the root node. Then, sum up the weighted values of edges to form the new graph. Determine the final minimum arborescence if no cycles exist in the new graph;If a ring exists in the new graph, shrink the ring to a point and change the edge weight. The procedure to change edge weights is as follows:(1)Choose a node (u) in the ring and set the incoming edge of this node as in[u] and outgoing edge as (u,i,w). i and w refer to the source node and weight, respectively;(2)Set the new edge weight of node u as (u,i,w−in[u]);(1)Return to Step 2 if the new weight graph contains rings;Expand the new graph if rings do not exist by the loop-breaking method (Hemminger, 1966; Gabow et al., 1986) [23,24]. The steps of the loop-breaking method are as follows:(1)Find a loop in the graph;(2)Remove the edge with the highest weight among the loops, but keep the graph connected;(3)Repeat this process until there are no loops in the graph (but they are still connected) and obtain the minimum spanning tree.

#### 2.1.4. Steps of the DPMFG Method

The network generated using the DMST method is extremely clear and easy to process, but too few edges are retained, and some critical information is frequently overlooked. To retain more edges, it is necessary to make the network as simple as possible while also ensuring that the retained information is sufficient for analysis of the characteristics of the network structure and to identify critical nodes in the network. Thus, we adopt the DPMFG method, which is similar to the DMST method. The calculation steps of the DPMFG method are as follows.

(1)By summing symmetric elements, convert the directed network into an undirected network;(2)Use the undirected PMFG to simplify a fully connected network to a network with only 3 (*N* − 2) edges remaining;(3)Restore each edge in the simplified undirected PMFG network to two directed edges, and keep the direction and weight of the side with the highest weight value as the edge and weight of the directed PMFG. Thus, the network is simplified to a directed PMFG network.

### 2.2. Topology Calculation and Community Partitioning Method for Multilayer Networks

#### 2.2.1. Centrality in Multilayer Networks

Unlike the aggregate algorithm, which simply aggregates the topology parameters from each single-layer network, the multilayer algorithm used in our study also considers connections between points from different layers. Domenico et al. (2013) [25] and Domenico et al. (2015) [26] used the tensorial formulation to calculate various multilayer network structure indicators.

When considering a multilayer network, a multilayer adjacency tensor can be used as an object for the complex relationship of the multilayer network. According to Domenico et al. (2013) [25], the multilayer network can be expressed as follows:(6)Mβδ˜αγ˜=∑i,j=1N∑h,k=1Lwij(hk)eα(i)eβ(j)eγ˜(h)eδ˜(k)
where N represents the number of nodes, L is the number of layers in the network, and wij(hk) denotes the connection between the node i of the layer h and node j of layer k. If it is a weighted network, w represents the corresponding weight value; otherwise, it is 1. If the network is a directed network, wij(hk) and wji(kh) are not necessarily equal. eα(i) denotes the αth component of the ith contravariant canonical vector (ei) in ℝN, and eβ(j) represents the βth component of the jth covariant canonical vector in ℝN. eγ˜(h) is the vector of the canonical basis in space ℝL, where the Greek index indicates the components of the vector, and the Latin index indicates the hth canonical vector. The multilayer adjacency tensor (Mβδ˜αγ˜) is a general object that can be used to represent the complicated relationships among nodes can be used to conveniently calculate the node centrality in a multilayer network.(1)Degree Centrality

The degrees of nodes are determined by the sum of each layer in the aggregated multilayer network calculation. In reality, the relationship between two nodes may differ between layers. Thus, when the degrees of each layer are directly summed, errors may occur. The same issue arises in the subsequent calculations of various centrality metrics. For multilayer networks, the association between different layers and whether the nodes between the layers can be connected need to be considered comprehensively. This method directly calculates various centrality measures as a whole, which can better reflect the actual relationship. Using the previously introduced multilayer adjacency tensor, we can directly calculate the multilayer degree centrality vector using the following formula:(7)$$K^{\alpha }=M_{\beta \tilde{\delta }}^{\alpha \tilde{\gamma }}U_{\tilde{\gamma }}^{\tilde{\delta }}u^{\beta }$$
where $U_{\tilde{\gamma }}^{\tilde{\delta }}$ denotes a second-order tensor, the elements of which are all equal to 1; and $u^{\beta }$ denotes the 1-vector, the components of which are all equal to 1. After some algebra, the multi-degree centrality vector can be written as:(8)$$K^{\alpha }=\sum_{h,k=1}^{L}k^{\alpha }(\tilde{h}\tilde{k})$$
where $k^{\alpha }(\tilde{h}\tilde{k})$ denotes the degree centrality vector that reflects the connections between layers $\tilde{h}$ and $\tilde{k}$. Clearly, the degree centrality vector calculated using this method differs from that calculated by aggregating the network layers into a single-layer network.

(2)Eigenvector Centrality

Eigenvector centrality is an indicator commonly used to describe the importance of a node that often has a high eigenvector centrality when its neighbors also have a high eigenvector centrality. The eigenvector centrality vector in a multilayer network can be used as the solution to the tensorial equation ($W_{j}^{i}v_{i}=\lambda_{1}v_{j}$), where $\lambda_{1}$ is the largest eigenvalue of $W_{j}^{i}$ (the adjacency tensor of the monoplex), and $v_{i}$ denotes the eigenvector centrality of node i. In the case of multilayer networks, we can obtain the eigenvector centrality by solving the tensorial equation:(9)$$M_{j\beta }^{i\alpha }\Theta_{i\alpha }=\lambda_{1}\Theta_{j\beta }$$
where $\lambda_{1}$ denotes the largest eigenvalue, and $\Theta_{i\alpha }$ is the corresponding eigentensor centrality of each node in each layer when accounting for the whole interconnected relationship. Thus, the multilayer Bonacich’s eigenvector centrality is given by $\Theta_{j\beta }=\lambda_{1}^{-1}M_{j\beta }^{i\alpha }\Theta_{i\alpha }$.

(3)PageRank Centrality

PageRank centrality quantifies how well a node is connected to other nodes. In general, if a node is connected to many other nodes, its PageRank value will be higher. The PageRank calculation simultaneously considers the PageRank value of the node connected to a node; if the PageRank value of the associated node is higher, the PageRank value of the considered node will also be higher. For the multilayer PageRank centrality calculation, we can start from the steady-state solution of the equation $p_{j\beta }(t+1)=R_{j\beta }^{i\alpha }p_{i\alpha }(t)$ in the case of a multilayer network, where $R_{j\beta }^{i\alpha }$ denotes the corresponding transition tensor that the walker jumps to a neighbor with rate r and teleports to any other node in the network with rate 1−r, and $p_{i\alpha }(t)$ represents the time-dependent tensor that described the probability of finding a walker at a particular node in a particular layer. This rank-4 tensor can be written as:(10)$$R_{j\beta }^{i\alpha }=rT_{j\beta }^{i\alpha }+\frac{\left(1-r\right)}{NL}u_{j\beta }^{i\alpha }$$
where $T_{j\beta }^{i\alpha }$ denotes the tensor of transition probabilities for jumping between pairs of nodes and switching between pairs of layers (the calculation process is detailed in Domenico et al. (2015) [27]), and $u_{j\beta }^{i\alpha }$ denotes the rank-4 tensor that with all components equal to 1. We can obtain the eigentensor ($\Omega_{i\alpha }$) of the transition tensor ($R_{j\beta }^{i\alpha }$); the multilayer PageRank is then calculated by simply contracting the layer index of the eigentensor with the 1-vector $\omega_{i}=\Omega_{i\alpha }u^{\alpha }$, i.e., by summing up across layers.

#### 2.2.2. Multiplex Infomap

Infomap is a stochastic and fast algorithm used to identify the best modular description of network flows according to the map equation, which measures the length required to communicate dynamics in a network and takes advantage of the information theory duality between finding regularities in data and compressing the data. Flow dynamics in multilayer networks are more complex than in single-layer networks because a random walker moves both intralayer and interlayer, and empirical interlayer weights are usually lacking. According to Daniel Edler et al. (2017) [28] and Domenico et al. (2015) [27], the multiplex map equation can be used to reveal the community structure of multilayer networks.

We use $W_{ij}^{\beta }$ to describe the intralayer adjacency of layer β and $D_{i}^{\alpha \beta }$ to describe the interlayer adjacency of physical node i. Then, we take the random walker moves according to intralayer links with probability 1−τ, and the random walker can move in different layers along any link of the physical node with probability τ. Therefore, the random walker switches from layer α to layer β with probability $s_{i}^{\beta }/S_{i}^{\alpha }$, where $S_{i}^{\alpha }=\sum_{\beta }D_{i}^{\alpha \beta }$ represents the total interlayer outlink weights of node i in layer α, and $s_{i}^{\beta }=\sum_{j}W_{ij}^{\beta }$ represents the total intralayer outlink weights of node i in layer β. Transition probabilities are used to describe the random walker dynamics:(11)$$P_{ij}^{\alpha \beta }(\tau )=(1-\tau )\delta_{\alpha \beta }\frac{W_{ij}^{\beta }}{s_{i}^{\beta }}+\tau \frac{W_{ij}^{\beta }}{S_{i}}$$
where $P_{ij}^{\alpha \beta }$ is the transition probability that node i in layer α moves to node j in layer β; τ is the probability that the random walker moves between layers; $s_{i}^{\beta }=\sum_{j}W_{ij}^{\beta }$ represents the total intralayer outlink weights in the same layer; $S_{i}=\sum_{\beta }s_{i}^{\beta }$ represents the total interlayer outlink weights across all layers; and $\delta_{\alpha \beta }$ is the Kronecker delta, which is 1 if α=β and 0 otherwise. $p_{i}^{\alpha }$ denotes the stationary distribution of state node i, α, i.e., node i in layer α, and can be derived from the recursive system of equations:(12)$$p_{i}^{\alpha }=\sum_{j,\beta }p_{j}^{\beta }P_{ji}^{\beta \alpha }$$

To ensure a unique ergodic solution in directed networks, we use teleportation at a rate κ to state nodes proportional to their intralayer in-link weights. To make the results more robust to the teleportation rate κ, we use unrecorded teleportation steps and recorded steps along links. First, we can obtain the recorded visit rates by calculating the stationary distribution with teleportation to state nodes proportional to their outlink weights.
(13)$${\tilde{p}}_{i}^{\alpha }=(1-\kappa )\sum_{j,\beta }p_{j}^{\beta }P_{ji}^{\beta \alpha }+\kappa \sum_{i,\alpha }\frac{S_{i}^{\alpha }}{S_{i}^{\alpha }}$$

We derive the recorded steps along links $q_{ji}^{\alpha \beta }$ and nodes $p_{i}^{\alpha }$ in next step:(14)$$q_{ji}^{\alpha \beta }={\tilde{p}}_{j}^{\beta }P_{ji}^{\beta \alpha }$$
(15)$$p_{i}^{\alpha }=\sum_{j,\beta }q_{ji}^{\alpha \beta }$$

For directed networks, the results are robust to the variation of teleportation rate κ in a wide range, and the results are independent of κ for undirected networks (Domenico et al., 2015) [27]. In the interest of simplicity, we use a teleportation rate of κ=0.15 throughout our directed networks.

The map equation can express the description length based on the rates at which a random walker enters and exits modules and visits nodes within modules, which can be used to calculate the rates for a multiplex network. $q_{\iota ↶}$ and $q_{\iota ↷}$ denote the transition rates at which the random walker enters and exits each module, respectively, where modules ι=1,2,…,m are assigned from a given partition (M) of state nodes i, α.
(16)$$q_{\iota ↶}=\sum_{\{i,\alpha \}\in J\neq \iota ,\{j,\beta \}\in \iota }q_{ij}^{\alpha \beta }$$
(17)$$q_{\iota ↷}=\sum_{\{i,\alpha \}\in \iota ,\{j,\beta \}\in J\neq \iota }q_{ij}^{\alpha \beta }$$

For module codebook ι, the physical node visit rates describe the random walker visiting each of the physical nodes in the module and can be written as:(18)$$p_{i\in \iota }=\sum_{\{i,\alpha \}\in \iota }p_{i}^{\alpha }$$

Now, we have expressed the description length of a random walker in terms of the three rates at which it enters and exits modules and visits state nodes of physical nodes within modules. Subsequently, we can obtain the two-level map equation in terms of multilayer networks with the per-step average description length (L(M)) of the trajectory of an ergodic random walker:(19)$$L(M)=q_{↶}H(\frac{q_{\iota ↶}}{q_{↶}})+\sum_{\iota =1}^{m}p_{\iota ↻}H(\frac{p_{i\in \iota }}{p_{\iota ↻}})$$
(20)$$p_{\iota ↻}=q_{\iota ↷}+p_{i\in \iota }$$
(21)$$q_{↶}=\sum_{\iota =1}^{m}q_{\iota ↶}$$
where H(•) denotes the Shannon entropy. For an identically distributed random variable (Z), $P(z_{i})$ represents the probability distribution of events ($z_{i}$), and Shannon entropy can be described as follows:(22)$$H(Z)=-\sum_{i}P(z_{i})\log_{2}P(z_{i})$$

Finally, we can use the Infomap search algorithm to acquire the optimal solution to minimize the previously introduced multiplex map equation.

### 2.3. Flowchart of the Methodologies

Section 2.1, Section 2.2 and Section 2.3 outline the main methods and steps used in the present study. These methods are mainly divided into two parts. In Section 2.1, we described the construction of multilayer networks, including the construction of fully connected networks (VD and TE methods) and two network-simplification methods (DMST and DPMFG methods).

In Section 2.2, we introduced the topology analysis method for multilayer networks, such as the topology calculation (PageRank, eigenvector, etc.) and community partitioning (Multiplex Infomap) methods.

The specific methodological steps can be represented by the following flowchart in Figure 1:

In the interest of simplicity, the formulae discussed in this article are summarized in Table 1 to clarify the calculation logic between the formulae.

Although the relationships between the formulae used in this article are listed in the table, these relationships are not intuitive, so we also present a diagram detailing the relationships between the formulae in Figure 2.

## 3. Data and Indicator Processing

### 3.1. Calculation of Comprehensive Index of Housing Prices

The sales price index of new commercial houses and secondhand houses is included in the housing price index data. The two indicators must be combined to create a comprehensive index. If the weight of the two indicators is arbitrarily chosen as a proportion, it is highly subjective and speculative. Thus, we employ the entropy method to objectively assign the two indicators. This method’s weight value is determined by the structural characteristics of the data. The greater the dispersion of an indicator, the more information it contains and the more weight it is given.

For a city, there is an m×n order data matrix $X={\left(x_{ij}\right)}_{m\times n}$, where m represents the length of time, and n is the number of indicators. The greater the difference between the index values of a column of data in a numerical matrix, the more information the index contains and the greater the role of the index is in the overall evaluation. For m months and n indicators of city r, $X_{\theta ij}$ denotes the jth indicator of city i in the θth year. According to Chen Minghua et al. (2020) [16] and Zhang Xiaoyan (2021) [29], the improved entropy weight method is used to calculate the weight of each indicator as follows.

(1)Because the order of magnitude of the indicators may differ, the data must be standardized before calculating the entropy value of each indicator. The indicators selected herein are all positive; therefore, the standardization formula for each indicator is:
(23)$$X_{\theta ij}^{\prime }=\frac{X_{\theta ij}-\min(X_{\theta j})}{\max(X_{\theta j})-\min(X_{\theta j})}$$
where $\max(X_{\theta j})$ is the maximum value of indicator j, and $\min(X_{\theta j})$ is the minimum value of indicator j;(2)To avoid 0 and negative values when calculating entropy, it is necessary to and add 0.1 to all values.
(24)$$X_{\theta ij}^{\prime \prime }=X_{\theta ij}^{\prime }+0.1$$(3)Determine the proportion of the jth index $P_{\theta ij}$ of each city in each year:
(25)$$P_{\theta ij}=\frac{X_{\theta ij}^{\prime \prime }}{\sum_{\theta =1}^{m}\sum_{i=1}^{r}X_{\theta ij}^{\prime \prime }}$$(4)Calculate the information entropy of the jth index. The lower the entropy value, the greater the difference between the indices. The information entropy is expressed as follows:
(26)$$e_{j}=-K\sum_{\theta =1}^{m}\sum_{i=1}^{r}P_{\theta ij}\ln P_{\theta ij}$$
where $K=\frac{1}{\ln(mr)}$.(5)Calculate the difference coefficient of the jth index:
(27)$$a_{j}=1-e_{j}$$(6)Calculate the weight of jth index:
(28)$$g_{j}=\frac{a_{j}}{\sum_{j=1}^{n}a_{j}}$$(7)Calculate the comprehensive index of housing price (ind). Because there is a minimal difference in the dimension of the sales price index between new commercial houses and the secondhand houses, the original data can be used to calculate the CIHP:
(29)$$ind_{\theta i}=\sum_{j=1}^{n}\left(g_{j}X_{\theta ij}\right)$$
where $ind_{\theta i}$ denotes the CIHP of city i in the θth year.

### 3.2. Analysis of the Comprehensive Index of Housing Prices

We select the capital cities of China’s provinces (Urumqi, Lhasa, and Xi’ning are excluded owing to a lack of data) and five well-developed cities (Shenzhen, Xiamen, Dalian, Ningbo, and Qingdao) as samples for our study. Therefore, a total of 33 cities are included for analysis; the time span runs from January 2012 to August 2021. The Wind database is used to calculate the housing price index. For subsequent research and analysis, we employ two methods to divide the selected cities. One method involves dividing them into eastern, central, and western regions according to their geographical location (eastern regions: Beijing, Tianjin, Shijiazhuang, Shenyang, Dalian, Shanghai, Nanjing, Hangzhou, Ningbo, Fuzhou, Xiamen, Jinan, Qingdao, Guangzhou, Shenzhen, Nanning, and Haikou; central regions: Taiyuan, Huhehaote, Changchun, Haerbin, Hefei, Nanchang, Zhengzhou, Wuhan, and Changsha; western regions: Chongqing, Chengdu, Guiyang, Kunming, Xi’an, Lanzhou, and Yinchuan); the method other entails dividing them into first-tier, new first-tier, second-tier, and third-tier cities according to the “Ranking of Cities’ Business Attractiveness in China 2021” (https://www.datayicai.com/readReport/267) (accessed on 27 January 2022) reported by the Yicai website (first-tier cities: Shanghai, Beijing, Shenzhen, and Guangzhou; new first-tier cities: Chengdu, Hangzhou, Chongqing, Xi’an, Wuhan, Nanjing, Tianjin, Zhengzhou, Changsha, Ningbo, Qingdao, and Shenyang; second-tier cities: Hefei, Kunming, Xiamen, Jinan, Fuzhou, Dalian, Haerbin, Changchun, Shijiazhuang, Nanning, Guiyang, Nanchang, Taiyuan, and Lanzhou; third-tier cities: Haikou, Hohhot, and Yinchuan). Because the selected regions are all in developed urban areas, the number of third-tier cities is relatively small.

To construct the CIHP for the 33 investigated cities, we employ the entropy method and use the average value of the CIHP during the study period for sorting and analysis. Table 2 displays the results.

Table 2 shows that, except for Hefei, Xiamen, and Nanning, the top ten cities in the CIHP are all first-tier or new first-tier cities. The top four cities, in particular, are all first-tier cities, namely Shenzhen, Guangzhou, Beijing, and Shanghai. However, there is no special urban-grade difference among the lower-ranked cities. Except for the first-tier cities, the other three types of cities are represented among the 10 cities with the lowest rankings.

From the perspective of regional distribution, except for Hefei, Wuhan, and Changsha in the central part of the CIHP, the other top 10 cities are all located in the eastern part of the country. The ten lowest-ranking cities have no discernible regional distribution and are located in the eastern, central, and western regions. This may be because the cities in our sample have high levels of economic development, and there is no obvious difference between the low-ranked cities. Consequently, there is little variation in the growth rate of housing prices, and the CIHP is also considerably similar.

The city with the lowest CIHP is Haikou. Nonetheless, its CHIP score is 102.3%, demonstrating that despite minor fluctuations in the growth of housing prices in the major cities in China, they all show an upward trend in general.

### 3.3. Selection and Source of Real Housing Price Data

The real housing price data used in our statical analysis were collected from the Anjuke website (https://www.anjuke.com/fangjia) (accessed on 2 December 2021). The selected cities and time intervals are consistent with the CIHP.

## 4. Results and Analysis

Based on the aforementioned methods and data, we determine the TE calculation bins, lags, and predictive horizon of GVD to be two, one (one month), and three periods (one quarter), respectively.

Subsequently, we use the DMST and DPMFG methods to translate the TE matrix and GVD matrix, which are generated by the CIHP and real housing prices, into two different four-layer networks. The characteristics of a multilayer network topology are used to describe the influence and contagion of housing prices in various locations.

We depict the multiplex networks of CIHP and the real housing price. To visualize the node cities with more edges in the complex network more intuitively, we use the node degree (the sum of outgoing and incoming degrees) as the node size. The larger the node, the more cities with which the city’s housing price is linked in the network. Generally, cities with more connecting nodes play an important role in the housing price association network. When a city’s housing prices are affected, the central city is not only more vulnerable to the impact of changes in housing prices in other cities, but changes in its housing prices can also spread to others cities.

### 4.1. Topology Analysis of Each Layer in Multilayer Complex Networks

The amount of information required to analyze central cities is insufficient if the number of edges in the simplified network obtained from the DMST is small and the degree difference of each node is not large. Hence, for the DMST-simplified network, we only examine the source nodes, that is, nodes with only outgoing degrees and no incoming degrees. To a considerable extent, the source nodes can represent the source of housing price correlation, which is of significance for analyzing the contagion source of housing price changes and preventing and controlling abnormal housing price changes. In other words, we analyze the source node of the DMST-generated simplified network and the center node of the DPMFG-generated simplified network.

The multilayer network structure of the CIHP and real housing prices can is shown in Figure 3.

The source cities of the CIHP DMST network are Guangzhou and Xiamen, whereas the source cities of the DMST network of real housing prices are Shenzhen and Guangzhou. Therefore, there are three source cities: Guangzhou, Xiamen, and Shenzhen. The CIHP in the source cities is high, particularly in Shenzhen and Guangzhou, which are the top two among all investigated cities. Xiamen is ranked slightly lower, in sixth place. These results show that the source of the contagion of housing price changes is often a city that has a rapidly rising housing prices, which plays a leading role for all cities. Thus, when housing prices in such cities rise or fall, the effect is distributed to other cities, causing housing prices in other cities to change.

The four-layer price complex networks simplified by the DPMFG show that the core cities differ depending on the complex networks. In the network constructed using the CIHP and TE methods, the top cities with the highest degree of nodes are Lanzhou, Changsha, Qingdao, and Wuhan. Only Shenzhen and Shanghai have slightly larger node sizes than the first top central cities among the four traditional first-tier cities. The top central cities are replaced by Shenyang, Tianjin, Hangzhou, and Haikou in the network built using the CIHP and GVD methods. The degree of the first-tier cities is higher than that determined using the TE method, although only a medium degree, with a large gap compared to the top cities.

From the perspective of the network built using real housing prices and the TE method, Qingdao, Harbin, and Yinchuan are the most prominent cities, ranking far higher than Beijing and Taiyuan. Only Beijing ranks in the middle of the first-tier cities, with the other three cities ranking extremely low. Zhengzhou and Xi’an rank far higher than other cities in the network constructed using real housing prices and the GVD method, whereas the first-tier cities rank lower.

Excluding some second-tier cities (Lanzhou and Harbin) and third-tier cities (Haikou and Yinchuan), the rest of the top-ranked cities are new first-tier cities. Qingdao, in particular, has a high degree in the two DPMFG-simplified networks built using the TE method, which is at the heart of the comparison. Conversely, Qingdao has a low degree for networks constructed using the GVD method. The traditional notion that first-tier cities should have a substantial impact on the housing prices of other cities is not reflected in the four DPMFG-simplified networks presented above. Except for Beijing and Shenzhen, which are in the middle of the network built using the TE method, the ranking in the other cases is extremely low.

The light- and dark-green stripes in Figure 4a,b represent the degree of DPMFG-simplified networks (CHPI networks and real housing price networks, respectively). Because the size of the selected node is calculated according to the degree, including both the part entering and leaving the node, it is impossible to tell whether the third-tier city is in the center because of its high housing price overflow or because it is considerably influenced by other cities. To further analyze the spillover effect of the central city and the extent of spillover from other cities, we calculate some structural indicators of the multilayer complex network to judge whether the central city has a large spillover effect or is affected by spillover from other cities.

### 4.2. Multilayer Network Centrality Index Analysis

We calculate the multi-PageRank centrality with reference to Domenico et al. (2015) [26] and the multi-degree and multi-eigenvector according to the work of Domenico et al. (2013) [25]. Furthermore, the aggregate value of the aforementioned indicators can be calculated according to Boccaletti S et al. (2014) [30]. The degree-in centrality and degree-out centrality can also be calculated for the degree centrality. Only the multi-centrality calculated by the multilayer algorithm will be used in the subsequent analysis, as it more accurately describes the structural characteristics of the multilayer network. Because multiple cities have the same value for some indicators, we use the same ranking value in these cases. The aforementioned indicators are calculated for the two four-layer networks of the CIHP and real housing prices, and the results are as follows:

Notably, the eigenvector centrality value for the network derived from the DMST is 0 because the calculation of the eigenvector requires each point to be reachable, whereas the DMST network cannot satisfy this condition (Figure 5c,d). Regarding the structural indicators of the CIHP, the cities with the highest multi-centrality are Tianjin, Changsha, and Shenyang, although this result may vary depending on multi-centrality. These three cities are among the top 10 multi-centrality cities listed earlier (Figure 4 and Figure 5) (except that Tianjin and Shenyang rank lower in multi-degree-out centrality (Figure 4e,f).

Although Tianjin ranks first in terms of the multi-centrality of real housing prices, Qingdao ranks first in multi-degree-out centrality, indicating that Qingdao has the characteristics of a highly central city (three types of multi-degree centrality are shown in Figure 4). Qingdao is also ranked highly in the CIHP, at least in the top 10, and in the top four for multi-degree and multi-degree-out centrality (Figure 4a,b,e,f). These results show that Qingdao has a considerably high position in terms of both complex network centrality calculated using the CIHP and direct centrality calculated using real housing price data. Therefore, it can be argued that Qingdao has a prominent position with respect to the contagion of housing price changes.

Regarding the comparison of Haikou and Yinchuan presented in the previous section, the multi-centrality indicators presented in Figure 4 and Figure 5 show that the centrality of the two cities ranks lower, possibly because they are highly ranked in some networks. Hence, to correctly evaluate whether a city is located in the center of the housing price correlation network, multiplex networks must be combined. It is inadvisable to rely solely on one or two complex networks to determine the status of a city in the correlation network.

Beijing ranked highest among the top 10 cities in terms of centrality of the four first-tier cities of Beijing, Shanghai, Guangzhou, and Shenzhen a total of six times, with the highest ranking of second in the multi-degree-out centrality of real housing prices (Figure 4f). Guangzhou only appeared in the top 10 once for the multi-degree-out centrality of the CIHP network (Figure 4e). Shanghai and Shenzhen were only in the top 10 three or four times, with a top ranking of third place. These findings also show that first-tier cities are not always at the center of the housing price correlation network. Tianjin, Qingdao, Shenyang, Beijing, and Taiyuan are the top 10 cities, with more than six occurrences; excluding Beijing and the second-tier city Taiyuan, the other three cities are new first-tier cities. These results also demonstrate that as the economy gradually develops, the dominance of the traditional first-tier cities with respect to housing price changes is gradually replaced by newly developed next-tier cities. Therefore, when controlling for the severe fluctuations in housing prices, we can focus on these new first-tier cities and conduct necessary housing price supervision to prevent rapid fluctuations in housing prices in a short period of time, which is not only conducive to the stability of housing prices in local cities but can also stabilize housing prices in other cities through the housing price association network.

Table 3 depicts the multi-centrality of the multilayer network of the CIHP and real housing prices. Here, “CIHP” represents the multi-centrality of the multilayer network constructed using the CIHP, and “price” represents the multi-centrality of the multilayer network constructed using the real housing prices. In addition, we count the number of times each city has been ranked in the top 10 in various multi-centrality categories, as shown in the last column of Table 3.

### 4.3. Community Analysis of Multilayer Networks

By employing the Multiplex Infomap method, we reveal the community structure of multilayer networks constructed using the CIHP and real housing prices. The Multiplex Infomap can integrate the characteristics of multiple networks and group individuals with similar characteristics into the same community, resulting in a large gap between different communities and a small gap between the same community. In this manner, we can investigate which cities have similar characteristics and gain an improved understanding of the linkage of housing prices, which is critical for subsequent housing price control. As shown in Figure 6, we use the Multiplex Infomap method to cluster the communities of the two four-layer networks and obtain a community clustering diagram of the CIHP and real housing prices.

According to the community clustering diagram of the CIHP, the cities can be divided into six communities. The first-tier cities of Beijing, Shanghai, Guangzhou, and Shenzhen are all part of the same community, as are Nanjing, Hefei, and Xiamen. These three cities are either adjacent to Shanghai (Nanjing and Hefei) or in the same coastal special economic zone as Shenzhen (Xiamen) and thus do not have a large gap relative to the first-tier cities. The first-tier cities are not all part of the same community according to the community clustering diagram of real housing prices. Except for Beijing and Guangzhou, which are in the same community, Shenzhen and Shanghai are in separate communities with other cities and are separated by a considerable distance. For example, Shanghai is in the same community as Hefei, Chengdu, Kunming, and Yinchuan, whereas Shenzhen is not clearly in the same community as the other cities. In general, the community clustering result obtained by using the CIHP is more effective for first-tier cities and is highly correlated with their actual economic level and geographical location.

The three cities of Tianjin, Qingdao, and Shenyang, which are among the top 10 cities in various categories of multi-centrality, are in the same community clustering of the CIHP, as are Jinan, Wuhan, and Changsha. Tianjin and Qingdao remain in the same community in terms of real housing price clustering, whereas Shenyang is a member of a separate community. Jinan and Wuhan are also in the same community as Tianjin and Qingdao, whereas Changsha is not in the same community as Tianjin or Shenyang. The clustering results of the two communities show that the central cities with a higher importance are more likely to share the same characteristics and attributes, making it easier for them to be grouped together.

It can be concluded that the multilayer complex network community clustering results obtained from the CIHP are superior; these results not only accurately divide the first-tier cities into the same community but also cluster the cities with higher rankings as per the multilayer network structure characteristics into the same community. There is still a substantial gap between the outcomes of real housing prices and the actual situation, and the outcomes of community division are less effective than those from the CIHP. Therefore, the multilayer complex network constructed using the CIHP is more consistent with the actual situation and has a high degree of recognition with respect to identifying important nodes.

## 5. Conclusions and Discussion

Focusing on the risk contagion of housing prices, we used the TE and GVD methods to construct a fully connected network with respect to housing price correlation. Subsequently, we simplified this network using the DMST and DPMFG, yielding two different four-layer multiplex network graphs of the CIHP and real housing prices, respectively. By analyzing the source cities from the DMST network, we found that the source cities are Guangzhou, Xiamen, and Shenzhen, which are all among the top cities in the CIHP. In addition, according to the node size of the DPMFG network, the central city is not a traditional first-tier city; rather, new first-tier cities appears more frequently in the central location.

Calculation the multi-centrality of multilayer networks shows that traditional first-tier cities do not rank extremely high in the various categories of multi-centrality. However, Tianjin, Qingdao, and Shenyang, which are new first-tier cities, occupy the central position in the multilayer networks. Furthermore, new first-tier cities replace first-tier cities as the core of the price fluctuation correlation network. Subsequently, we used the Multiplex Infomap method to cluster multilayer networks and found that the community cluster based on the CIHP is most effective. The first-tier cities and some cities near Shanghai are members of the same community, as are Tianjin, Qingdao, and Shenyang, with high network structure indicators.

The methods of building multilayer networks and dynamic topology extraction can also be applied to financial networks. For example, when building stock/bank multilayer networks, we can use the closing prices of multiple stocks as the research object, build a multilayer financial network through VD and TE methods, and build dynamic financial networks for crisis early warning research. The method used in the present study has good applicability in the financial field.

In the future, we will consider using machine learning to identify early warning signs of rising housing prices. We intend to convert housing data into a symbolic sequence (0 and 1) corresponding to price rises and falls. Through the sliding window method and multilayer network structured indicators (multi-PageRank, multi-eigenvector, etc.) constructed above as the characteristics (X1, X2, X3, etc.) of a machine learning model, various machine learning models can be used to forecast the rise or fall in housing prices.

## Figures and Tables

**Figure 1 entropy-24-01305-f001:**
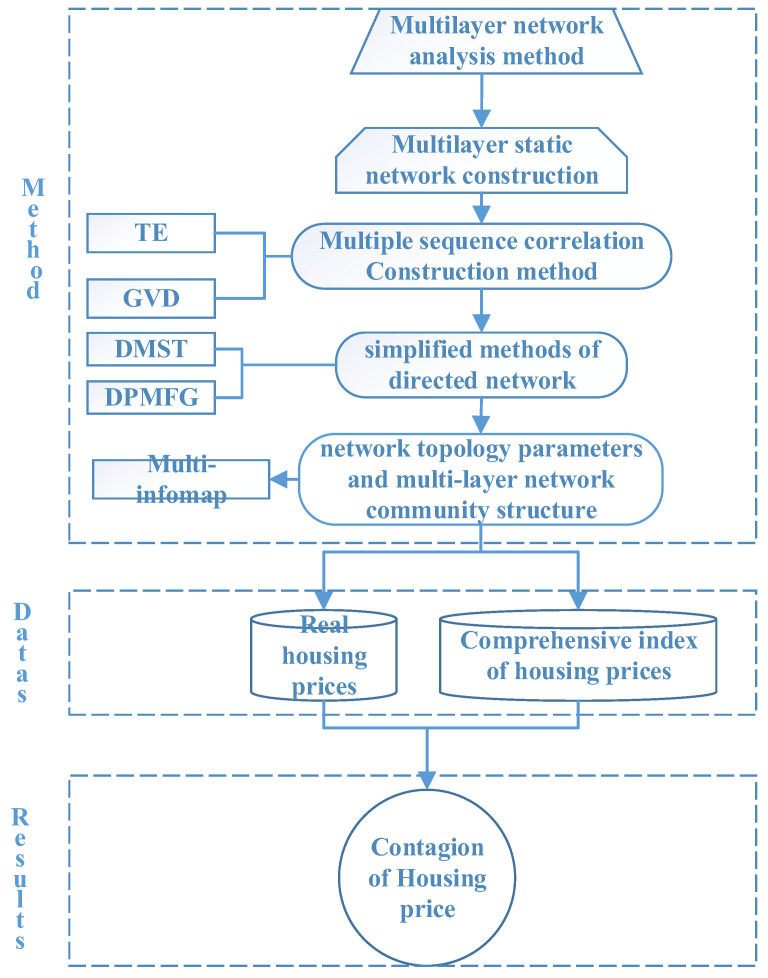
Empirical analysis flowchart.

**Figure 2 entropy-24-01305-f002:**
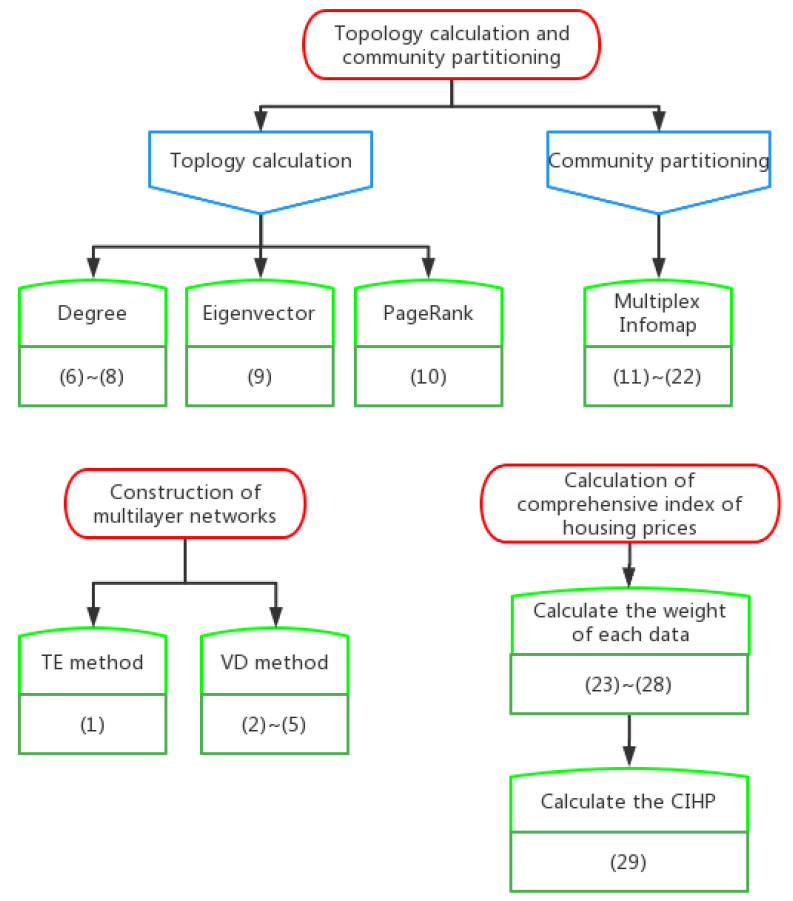
Diagram of the relationships between formulae.

**Figure 3 entropy-24-01305-f003:**
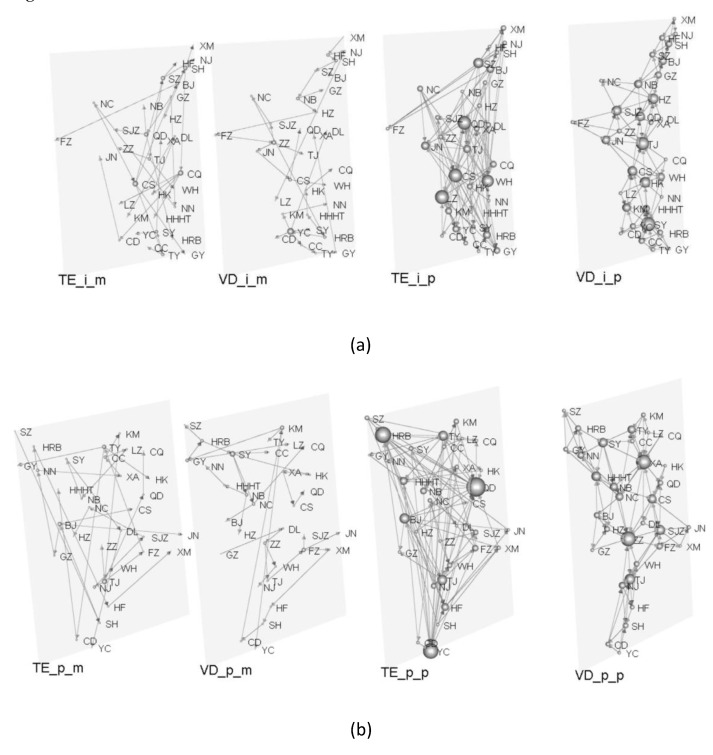
Multilayer complex network diagram of the comprehensive index of housing prices and real housing prices. (**a**) The network is built on the comprehensive index of housing prices. From left to right, the first figure depicts a network constructed using TE and DMST methods, the second depicts a network constructed using VD and DMST methods, the third depicts a network built using TE and DPMFG methods, and the fourth depicts a network constructed using VD and DPMFG methods. (**b**) Network built using real-world housing price data. The first figure depicts a network constructed using TE and DMST methods, the second depicts a network built using VD and DMST methods, the third depicts a network constructed using TE and DPMFG methods, and the fourth depicts a network built using VD and DPMFG methods.

**Figure 4 entropy-24-01305-f004:**
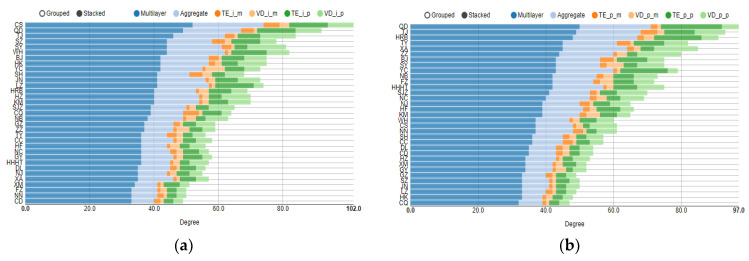
The multi-degree, multi-degree-in and multi-degree-out centrality of multilayer networks. (**a**) The multi-degree centrality of CIHP networks; (**b**) the multi-degree centrality of real housing price networks; (**c**) the multi-degree-in centrality of CIHP networks; (**d**) the multi-degree-in centrality of real housing price networks; (**e**) the multi-degree-out centrality of CIHP networks; (**f**) the multi-degree-out centrality of real housing price networks.

**Figure 5 entropy-24-01305-f005:**
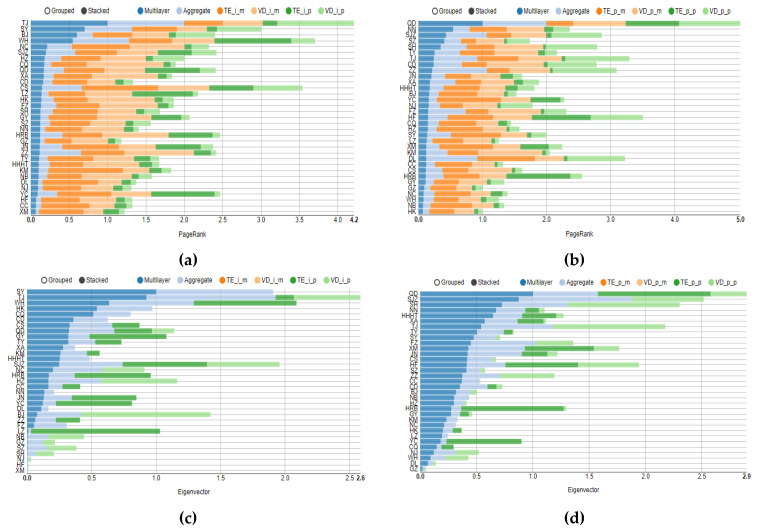
The multi-PageRank and multi-eigenvector centrality of multilayer networks. (**a**) The multi-PageRank centrality of CIHP networks; (**b**) the multi-PageRank centrality of real housing price networks; (**c**) the multi-eigenvector centrality of CIHP networks; (**d**) The multi-eigenvector centrality of real housing price networks.

**Figure 6 entropy-24-01305-f006:**
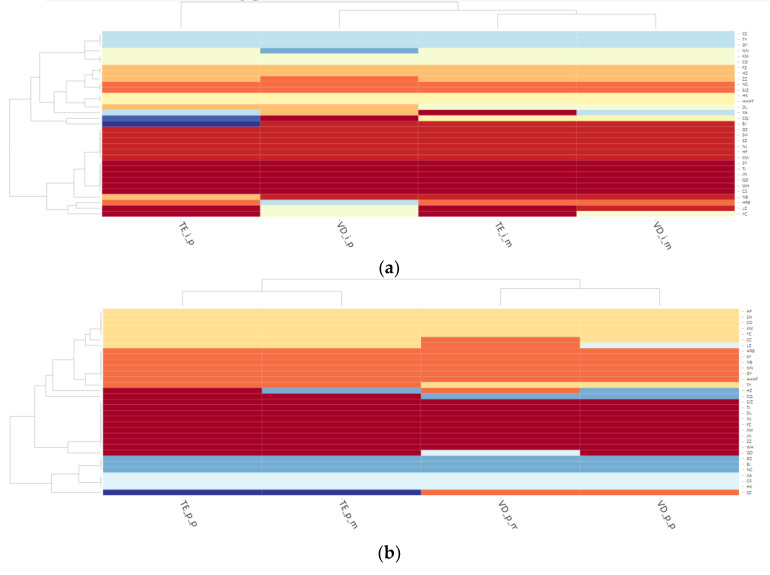
Community clustering graph of multilayer networks. (**a**) Community clustering of multilayer networks using the comprehensive index of housing prices. (**b**) Community clustering of multilayer networks using real housing prices.

**Table 1 entropy-24-01305-t001:** Compilation of the formulae discussed in this article.

Description	Formula Number	References
2.1 Construction of multilayer networks		
Network constructed by TE method	(1)	Schreiber (2000) [17], Chen et al. (2014) [18]
Network constructed by VD method	(2)~(5)	Diebold and Yilmaz (2014) [19]
2.2 Topology calculation and community partitioning method for multilayer networks		
Degree centrality	(6)~(8)	Domenico et al. (2013) [25], Domenico et al. (2015) [26]
Eigenvector centrality	(9)	
PageRank centrality	(10)	
Multiplex Infomap	(11)~(22)	Domenico et al. (2015) [27], Daniel Edler et al. (2017) [28],
3.1 Calculation of housing price linkage index		
Calculate the weight of each data point	(23)~(28)	Chen Minghua et al. (2020) [16], Zhang Xiaoyan (2021) [29]
Calculate the comprehensive index of housing price	(29)	

**Table 2 entropy-24-01305-t002:** Cities ranked according to the comprehensive index of housing prices.

Ranking	City	Abbreviation	CIHP
1	Shenzhen	SZ	110.6588
2	Guangzhou	GZ	107.6885
3	Beijing	BJ	107.54
4	Shanghai	SH	107.1897
5	Heifei	HF	106.8774
6	Xiamen	XM	106.8587
7	Nanjing	NJ	106.3807
8	Wuhan	WH	105.8394
9	Nanning	NN	104.798
10	Changsha	CS	104.6763
11	Xi’an	XA	104.6693
12	Kunming	KM	104.5756
13	Zhengzhou	ZZ	104.5389
14	Fuzhou	FZ	104.5032
15	Shenyang	SY	104.4324
16	Hangzhou	HZ	104.3276
17	Huhehaote	HHHT	104.1595
18	Shijiazhuang	SJZ	103.9711
19	Yinchuan	YC	103.8835
20	Chongqing	CQ	103.8156
21	Guiyang	GY	103.7865
22	Tianjin	TJ	103.7282
23	Nanchang	NC	103.7266
24	Haerbin	HRB	103.6739
25	Dalian	DL	103.6114
26	Ningbo	NB	103.4872
27	Taiyuan	TY	103.4435
28	Chengdu	CD	103.4259
29	Jinan	JN	103.4219
30	Changchun	CC	103.2341
31	Lanzhou	LZ	102.9835
32	Qingdao	QD	102.8244
33	Haikou	HK	102.3365

**Table 3 entropy-24-01305-t003:** Multi-centrality ranking of cities.

City	PageRank		Eigenvector		Degree		Degree-In		Degree-Out		Rank in Top 10
	CIHP	Price	CIHP	Price	CIHP	Price	CIHP	Price	CIHP	Price	
TJ	1	7	2	7	3	2	1	8	22	1	9
QD	9	1	8	1	2	1	4	1	2	16	9
SY	2	20	1	9	4	7	3	18	22	2	7
BJ	3	13	23	19	7	7	9	18	10	2	6
TY	24	6	10	8	21	4	18	9	19	5	6
SJZ	6	3	14	2	16	13	7	5	28	29	5
WH	4	31	3	31	4	18	2	13	28	16	4
XA	10	11	11	6	27	4	26	4	15	12	4
CS	12	26	7	13	1	18	7	13	1	16	4
SH	16	5	30	3	10	21	18	15	5	16	4
ZZ	23	9	24	16	19	6	18	6	15	9	4
YC	30	14	21	28	7	7	11	3	9	33	4
NC	5	30	15	25	21	14	26	10	11	10	3
CQ	8	18	5	29	16	33	32	26	2	29	3
HK	14	33	4	26	7	28	15	32	5	16	3
FZ	15	16	25	10	31	10	18	18	33	4	3
SZ	18	4	29	15	4	28	15	26	2	22	3
NN	19	2	19	4	31	18	30	10	19	22	3
HRB	20	27	16	22	13	3	9	2	15	12	3
JN	22	10	20	12	10	28	4	18	22	29	3
HHHT	25	12	13	5	21	10	11	6	28	12	3
NB	27	32	27	20	18	10	30	15	5	5	3
CD	11	8	6	18	31	23	25	18	28	16	2
LZ	13	21	26	27	10	28	4	32	22	16	2
HZ	7	19	17	21	13	25	11	18	11	22	1
GY	17	28	9	23	21	25	15	26	22	16	1
GZ	21	29	28	33	19	28	32	26	5	22	1
KM	26	23	12	24	13	15	11	26	11	7	1
NJ	29	15	31	30	27	15	18	18	22	8	1
HF	31	17	32	14	21	15	26	10	11	12	1
DL	28	24	22	32	27	23	26	18	15	16	0
CC	32	25	18	17	21	21	18	26	19	10	0
XM	33	22	33	11	30	25	18	15	28	29	0

## Data Availability

Data available in a publicly accessible repository. (https://www.anjuke.com/fangjia) (accessed on 27 June 2022).

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
