# Peer review of "Research on China’s Risk of Housing Price Contagion Based on Multilayer Networks"

_entropy, 2022, doi:10.3390/e24091305_

Round 1
Reviewer 1 Report
Summary
This paper analyses a study of housing price developments in various cities in China. Its fundamental thesis focuses on the analysis of the diffusion of price changes across cities classified according to the categories of first-tier cities and new first-tier cities and so on. To carry out this study, they perform network analysis, focusing on transfer entropy, and develop a housing price linkage index (HPLI) for the period 2012-2021.
Comments
This paper is very interesting and provides a very good analysis of the linkages between housing price structures in different cities in China.
The tools used are appropriate and are used in the right way, allowing us to obtain several conclusions about the analyzed market. The authors consider a network analysis as the basis for the study and apply several tools: transfer entropy, generalized variance decomposition, directed minimum spanning tree and directed planar maximally filtered graph methods. The final result reveals a linkage structure of the price evolution between the different cities according to their classification.
At the same time, this analysis, which is correct but only descriptive, is accompanied by a predictive study which seeks to analyze the possible evolution of prices through the formulation of an index (HPLI) which makes it possible to anticipate price variations in some cities according to the variations observed in others.
The work is very good and can be published in its current version. Some minor corrections of formal aspects would be necessary, which do not affect the content or the quality of the work done.
In the future, it would be interesting to consider not only the evolution of housing prices through the interrelationship between cities but also considering some variables that affect these prices: population evolution, city structure, geographic location, etc.
Specific Comments
Some specifics questions to consider:
1. Page 4 (Lines 127-132) Paragraph repeated (lines 133-138)
2. Page 11 (Line 346). Re-do expression in this line.
3. Figures. The quality of figures must be improved to visualize them correctly.
4. Page 21 (Line 580). Review the title and description of Figure 3. It could be wrong.
5. Page 31 (Lines 867-870). References numbered (30) and (31) are not ordered, they might be numbers (23) and (24).
Finally, the quality of typos and equations must be improved and use the same font for the whole paper.
Author Response
- Page 4 (Lines 127-132) Paragraph repeated (lines 133-138).
Thanks for your suggestion. Duplicate parts in the text have been removed.
- Page 11 (Line 346). Re-do expression in this line.
Thanks for your suggestion. The formula has been rewritten and the symbols involved in the formula are explained.
- The quality of figures must be improved to visualize them correctly.
Thanks for your suggestion. Figure 2 has been redrawn(Figure 3 in the new manuscript ) and Figure 3(Figure 4 and 5 in the new manuscript ) has been divided into two figures to improve the quality.
- Page 21 (Line 580). Review the title and description of Figure 3. It could be wrong.
Thanks for your suggestion. The description of Figure 3(Figure4 and Figure5 in the new manuscript) has been completed, such as the method and the centrality type of the graph.
- Page 31 (Lines 867-870). References numbered (30) and (31) are not ordered, they might be numbers (23) and (24).
Thanks for your suggestion. The sequence number of reference (23), (24), (30) and (31) has been adjusted.
Reviewer 2 Report
-
The Authors presented very interesting research relevant to development of the China's housing price risks contagion and early warning based on multi-layer network.
Scope of the research is up-to-date and vital in wider international discussions. The paper generally was elaborated comprehensively, however it is not enough clear for readers and needs some improvements.
Below a few major comments to the specific assumptions and results:
- the authors didn’t constitute the scientific thesis or hypothesis,
- despite the fact that the methodology of the proposed solution related to finding housing price correlation is comprehensive and detailed, it is difficult to find in it universality that could be applied in other markets due to the fact that there is no sequential-procedure provided that could play a leading role,
- the article should be rewritten because of the difficulty in reading the successive formulas, by first introducing a diagram with cross-references to the formulas introduced in terms of their cause and effect.
- figure 2 and 3 are illegible,
- table 4 – how exactly the advance time of peak and though was calculated? ..it is average measure? IF yes than average measure is not the reliable information in this specific issue.
- in that kind of analyses very crucial point is assumption of sliding window that in this study was - 12-month window and 1-month step, this assumption should be supported by the initial trend analyses
- in the paper there is the lack of disadvantages of applied method discussion (due to the fact that all methods and models are just try to imitate the realty, the drawbacks should be provided ,
- in final statement the authors claim that “in the future, will consider using machine learning
to predict the HPLI and identify early warning signs for rising housing prices” – the question for which exactly data and their form they are going to use machine learning method and real reason for that.The paper needs significant improvement.
Author Response
- The authors didn’t constitute the scientific thesis or hypothesis.
Thanks for your suggestion. In the new manuscript, we have added the summary of methods and processes to constitute a complete methodology before the flowchart of this article.
- Despite the fact that the methodology of the proposed solution related to finding housing price correlation is comprehensive and detailed, it is difficult to find in it universality that could be applied in other markets due to the fact that there is no sequential-procedure provided that could play a leading role.
Thanks for your suggestion. It is similar to the question1, using our complete set of methodology, we can also utilize several stock closing prices instead of housing prices.
Firstly, we can create financial multilayer networks and analyze network topology of the multilayer networks. Secondly, we can use the CBEDETE method to create dynamic financial multilayer networks and extract network topologies to construct stock related index series. Finally, we can utilize the Gaussian smoothing method to polish the stock related index series and extract peak as the early warning signals of financial crises.
Details have been written in Section 5 of this article.
- The article should be rewritten because of the difficulty in reading the successive formulas, by first introducing a diagram with cross-references to the formulas introduced in terms of their cause and effect.
Thanks for your suggestion. We have made a table to arrange the formulas and the corresponding references. The details are shown in Table 2.
- Figure 2 and 3 are illegible.
Thanks for your suggestion. Figure 2 and 3 have been redrawn.
- Table 4 – how exactly the advance time of peak and though was calculated? ..it is average measure? IF yes than average measure is not the reliable information in this specific issue.
Thanks for your suggestion. The time position of the peak/trough is extracted from the smoothed series (by using the Gaussian smoothing method(Lu Qiu and Huijie Yang, 2020)), not the average housing prices of each district in a city. The advance time is the time interval between the time when peak/trough of the smoothed series occur and the time when the tendency of the real housing price starts to change.
- In that kind of analyses very crucial point is assumption of sliding window that in this study was - 12-month window and 1-month step, this assumption should be supported by the initial trend analyses.
Thanks for your suggestion. The window length in our works is set as 12 months (exactly one year) because of the limitation of the housing price data volume and length, and we take 1 month as the step to get the most consecutive segments.
- In the paper there is the lack of disadvantages of applied method discussion (due to the fact that all methods and models are just try to imitate the realty, the drawbacks should be provided.
Thanks for your suggestion. The description of the theory has been answered in answer 1. The drawbacks of applied methods in this paper are written in the conclusion and discussion part.
- In final statement the authors claim that“in the future, will consider using machine learning
to predict the HPLI and identify early warning signs for rising housing prices” – the question for which exactly data and their form they are going to use machine learning method and real reason for that.
Thanks for your suggestion. Due to the unsupervised characteristics of our method in this paper. We will intend to convert housing data into a symbolic sequence (0 and 1) through price rises and falls. Symbolic sequences can be regarded as target variables in machine learning models.
We can use various topologic sequences (degree, mediation coefficient, clustering coefficient, etc.) in the constructed dynamic short sequence network as characteristic variables (X1, X2, X3, etc.) to forecast the rise or fall of housing prices.
Reviewer 3 Report
It appears impossible to constructively evaluate this material. It contains a large collection of loosely related catchy slogans mixed up in the context of housing prices. The notation introduced is not consistent and the formulae written down are not properly explained nor derived. As a result the degree of scientific rigour is here none. The numerical results shown are unreproducible. The main conclusion that the introduced HPLI index provides an early warning signs for rising housing prices, based on the selected examples of Fig.5, is unreliable and can only mislead an uninformed reader.
Author Response
First of all, thank you very much for your valuable comments. Our research mainly focuses on empirical research. The calculation of transfer entropy, variance decomposition, and topological index in multilayer networks are based on authoritative literature, and the derivation of the formula is not involved. The data we utilize is official data that is publicly accessible, so the results are reproducible. For Figure 5(Figure 7 in the revised text), we applied a new method for short-sequence transfer entropy calculation(CBEDE) to describe the relationship between cities, and to construct a dynamic network sequence. We construct the HPLI index and use Gaussian smoothing method to process the index sequence, and from the 6 city results in this paper, we get consistent results (the peaks of the HPLI series frequently appear before a sharp rise in housing prices, while the troughs of the HPLI are followed by a plunge in housing prices). Therefore, we deem that our findings have some credibility.
Round 2
Reviewer 1 Report
The revised version of the paper is almost correct and could be accepted and published. Before publication, some text editing corrections must be done.
Typos (fonts and dimensions) in equations must be unified, i.e. Page 11 - line 342, an illegible equation is shown.
Correct the number of figures (There are two figures with the number 5 and no figure with the number 4).
Author Response
- Typos (fonts and dimensions) in equations must be unified, i.e. Page 11 - line 342, an illegible equation is shown.
Thanks for your suggestion. The equation has been rewritten.
- Correct the number of figures (There are two figures with the number 5 and no figure with the number 4).
Thanks for your suggestion. The number of figures have been corrected.
Reviewer 2 Report
- Reviewer comments in red
- The authors didn’t constitute the scientific thesis or hypothesis.
Thanks for your suggestion. In the new manuscript, we have added the summary of methods and processes to constitute a complete methodology before the flowchart of this article.
This is not answer for the problem
- Despite the fact that the methodology of the proposed solution related to finding housing price correlation is comprehensive and detailed, it is difficult to find in it universality that could be applied in other markets due to the fact that there is no sequential-procedure provided that could play a leading role.
Thanks for your suggestion. It is similar to the question1, using our complete set of methodology, we can also utilize several stock closing prices instead of housing prices.
Firstly, we can create financial multilayer networks and analyze network topology of the multilayer networks. Secondly, we can use the CBEDETE method to create dynamic financial multilayer networks and extract network topologies to construct stock related index series. Finally, we can utilize the Gaussian smoothing method to polish the stock related index series and extract peak as the early warning signals of financial crises.
Details have been written in Section 5 of this article.
Not answer for the question.
- The article should be rewritten because of the difficulty in reading the successive formulas, by first introducing a diagram with cross-references to the formulas introduced in terms of their cause and effect.
Thanks for your suggestion. We have made a table to arrange the formulas and the corresponding references. The details are shown in Table 2.
Author/s respond partially. The contents of table 2 is insufficient.
- Figure 2 and 3 are illegible.
ok
Thanks for your suggestion. Figure 2 and 3 have been redrawn.
- Table 4 – how exactly the advance time of peak and though was calculated? ..it is average measure? IF yes than average measure is not the reliable information in this specific issue.
Thanks for your suggestion. The time position of the peak/trough is extracted from the smoothed series (by using the Gaussian smoothing method(Lu Qiu and Huijie Yang, 2020)), not the average housing prices of each district in a city. The advance time is the time interval between the time when peak/trough of the smoothed series occur and the time when the tendency of the real housing price starts to change.
ok
- In that kind of analyses very crucial point is assumption of sliding window that in this study was - 12-month window and 1-month step, this assumption should be supported by the initial trend analyses.
Thanks for your suggestion. The window length in our works is set as 12 months (exactly one year) because of the limitation of the housing price data volume and length, and we take 1 month as the step to get the most consecutive segments.
ok
- In the paper there is the lack of disadvantages of applied method discussion (due to the fact that all methods and models are just try to imitate the realty, the drawbacks should be provided.
Thanks for your suggestion. The description of the theory has been answered in answer 1. The drawbacks of applied methods in this paper are written in the conclusion and discussion part.
The paper needs sufficiently prepared method analyses (procedure) according to their possibility of use and drawbacks.
- In final statement the authors claim that“in the future, will consider using machine learning
to predict the HPLI and identify early warning signs for rising housing prices” – the question for which exactly data and their form they are going to use machine learning method and real reason for that.
Thanks for your suggestion. Due to the unsupervised characteristics of our method in this paper. We will intend to convert housing data into a symbolic sequence (0 and 1) through price rises and falls. Symbolic sequences can be regarded as target variables in machine learning models.
We can use various topologic sequences (degree, mediation coefficient, clustering coefficient, etc.) in the constructed dynamic short sequence network as characteristic variables (X1, X2, X3, etc.) to forecast the rise or fall of housing prices.
What will this kind of solution bring new and when is it necessary
Author Response
Thanks for your suggestion. Please find the attached answer for questions.

Reviewer 3 Report
By stating that the results are unreproducible I didn't mean that the initial data are unavailable but that the procedure presented is totally misleading, confusing and very likely wrong in several aspects. Just some examples: (i) setting the diagonal elements to 0 (thus setting its trace to 0) as in (12) changes the character of the matrix and makes things uncontrolable, (ii) arbitrarily adding up 0.1 (why just such a value?) in (33) is completely unjustfied (and changes the value of entropy) especially that by (32) the corresponding values (X) are already nonegative, (iii) what (31) expresses is totally unclear and looks just postulated and not derived, etc, etc...
Author Response
By stating that the results are unreproducible I didn't mean that the initial data are unavailable but that the procedure presented is totally misleading, confusing and very likely wrong in several aspects. Just some examples:
1 setting the diagonal elements to 0 (thus setting its trace to 0) as in (12) changes the character of the matrix and makes things uncontrolable,
We set the diagonal to 0, which means that the information transmission from the node to itself is 0, that is, to remove the self loop during visualization.
(ii) arbitrarily adding up 0.1 (why just such a value?) in (33) is completely unjustfied (and changes the value of entropy) especially that by (32) the corresponding values (X) are already nonegative,
We normalize the raw data in Eq. (32) so that the data is distributed between 0 and 1. ​Nevertheless, it is inevitable that when the original data is exactly the minimum value, the corresponding processed data will be 0, so this paper adds 0.1 to the standardized data uniformly to make the data all positive. Although the magnitude of the data processed above has changed, the overall data change trend is unchanged, so the entropy calculated by formula (32) and formula (33) will not change greatly. The calculation of the comprehensive index of housing prices is referring to Chen Minghua et al.(2020) and Zhang Xiaoyan(2021).
(iii) what (31) expresses is totally unclear and looks just postulated and not derived, etc, etc...
The (31) in this paper is expression of the CBEDETE (TE using the correlation-dependent balanced estimation of diffusion entropy) method whose effectiveness have been verified in our published article (Lu Qiu and Huijie Yang (2020)).
Round 3
Reviewer 2 Report
The paper can be published. All amendments were provided.
Reviewer 3 Report
The manuscript still does not contain any significant modifications that wouldallow me to change my previous recommendation.
A scientifically rigorous response to my remarks is avoided.
